# “I Sometimes Ask Patients to Consider Spiritual Care”: Health Literacy and Culture in Mental Health Nursing Practice

**DOI:** 10.3390/ijerph16193589

**Published:** 2019-09-25

**Authors:** Adwoa Owusuaa Koduah, Angela Y.M. Leung, Doris Y.L. Leung, Justina Y.W. Liu

**Affiliations:** Centre of Gerontological Nursing, School of Nursing, The Hong Kong Polytechnic University, Kowloon, Hung Hom, Hong Kong; owusuaa.koduah@connect.polyu.hk (A.O.K.);

**Keywords:** health literacy, mental health nursing, culture, beliefs, Ghana

## Abstract

While health literacy influences better outcomes of mental health patients, sociocultural factors shape the nature of the relationship. On this matter, little is known about how sociocultural factors affect health literacy practices of nurses, especially in low-income countries. This paper examines how local precepts, within culture and language, shape mental health nurses’ (MHNs) practice and understanding of patients’ health literacy level in Ghana. The study used a qualitative descriptive design involving 43 MHNs from two psychiatric hospitals. Conventional content analysis was used to analyze the data. Although the MHNs acknowledged the importance of health literacy associated with patients’ health outcomes, their practice was strongly attributed to patients’ substantial reliance on cultural practices and beliefs that led to misinterpretation and non- compliance to treatments. MHNs shared similar sociocultural ideas with patients and admitted that these directed their health literacy practice. Additionally, numerous health system barriers influenced the adoption of health literacy screening tools, as well as the MHNs’ low health literacy skills. These findings suggest MHNs’ direct attention to the broader social determinants of health to enhance the understanding of culture and its impact on health literacy practice.

## 1. Introduction

Mental health is a major concern in high, low-, and middle-income countries. Mental disorders account for an estimated 12% of the global burden of diseases. Further, mental disorders create one of the most unrecognized sources of health care disparities [1,2]. For instance, people diagnosed with a mental illness are at higher risk of developing preventable chronic diseases and have the highest rates of mortality [2,3]. 

In Ghana, approximately 2.4 million people have mental disorders [4]. Despite the high incidence, a treatment gap exists for severe mental disorders, estimated to be as high as 95%. Moreover, the gap is wider among persons suffering from mild to moderate mental health problems [1,4]. In part, this is a result of gaps in mental health education in Ghana generating low health literacy; exacerbated by the lack of its interpretation from cultural and religious perspectives [5]. 

Many Ghanaians firmly believe that mental illness is caused by the existence and manipulation of witches, ancestral spirits, sorcerers, and other demonic forces [6]. As a result, in Sub-Sahara Africa, mental health issues are often too sensitive to discuss with patients and their families due to the shame and stigma associated with them [5]. This context affects how health professionals deliver information and how patients and their relatives interpret it. Yet little is known about how such sociocultural factors influence nurses’ health literacy practices, especially in low-income countries. This study argues that health literacy and its practice may be a useful framework to examine and understand the nature of disparities in mental healthcare [2,7,8]. 

### Background

Health literacy is the degree to which an individual can obtain, process, and understand basic health information and the services needed to make appropriate health decisions [9]. Research indicates that although health literacy affects mental health outcomes, sociocultural factors determine the nature of the relationship. Health literacy practice refers to patient-centered protocols and strategies that involve the assessment of patients’ health literacy and the actions taken to minimize the negative consequences of low health literacy [10]. Health literacy practice goes beyond the individual and incorporates the broader health system, including health professionals and the logistics of the local culture, the education system, and how these factors interact [11]. 

Common approaches in health literacy practice include: use of standardized health communication tools, using of plain language, face-to-face communication and drawings, involving frontline staff (for example nurses, doctors, administrators), partnering with patients to improve communication, and establishing a health literate organization, which develops universal precautions for clear communication [10,12]. In this regard, health literacy practice is a promising approach to improving public and patients’ health outcomes through knowledge and application of health information [13,14].

Empirical data suggests that nurses do not routinely adopt available health literacy practices [12,15]. Nurses are often not well-equipped to face and address the challenges of low health literacy within clinical settings [16,17]. This is due to the fact that nurses have not routinely received health literacy education as part of their professional preparation [18,19] Therefore, while low health literacy of mentally-ill patients prolongs care and recovery, lack of adequate health literacy skills of mental health nurses (MHNs) is thought to exacerbate the problem [7]. 

Ghana is a heterogeneous nation located in Sub-Sahara Africa [6,20]. The country has over 100 culturally distinct groups of people with diverse linguistic (50 to 100 languages and dialects spoken by the various ethnic groups), spiritual, and religious backgrounds [6,21]. English is the official language in Ghana and it is used universally at all levels of the educational system [4]. The literacy rate of Ghana as of 2010 was 71% [22].

It is estimated that about 71.2% of Ghana’s population are Christians, 17.6% share the Islamic faith, and 5.2% associate with the Ghanaian Traditional religion [23]. However, a significant number of the population practice traditional beliefs alongside Christianity or Islam. Traditional beliefs form a significant part of Ghanaian culture and everyday normative practices [23]. This multi-religious practice encourages pluralistic health-seeking behaviors among the public. People commonly patronize two or more of the services of orthodox hospitals, prayer camps, herbalists, and traditional healers when they are sick [5]. For mental health problems, about 70% of the families of mental health patients prefer herbal and/or spiritual treatment to orthodox services [24]. This has been attributed to the fact that families of mentally ill patients and traditional and faith healers have shared understanding and beliefs of mental illness [5,25].

MHNs’ health literacy practice is even more complicated when one considers the crucial role of cultural beliefs and practices in the treatment of mental illness in low socioeconomic places, such as Ghana. This paper examines how contextual precepts such as culture, religious beliefs, and language situate mental health nurses’ (MHNs) practice and understanding of patients’ health literacy in Ghana. 

## 2. Methods 

### 2.1. Study Design and Setting 

The study used a qualitative descriptive design involving 43 MHNs from two psychiatric hospitals. The two hospitals were chosen as they provide services to the majority of mentally-ill persons and employ the most of MHNs in the country. A paradigm of post-positivism was adopted [26,27] to examine MHNs’ health literacy knowledge, practices and the influence of culture and social and institutional factors. This paradigm and research design provided an appropriate strategy to broadly explore a topic where minimal studies exist, with regard to describing “perceptions, inclinations, sensitivities, and sensibilities” of a localized phenomenon [28]. 

### 2.2. Sampling and Data Collection 

The study adopted a purposive sampling method [29], to recruit registered and permanently employed MHNs working in two government teaching psychiatric hospitals in Ghana. Inclusion criteria included registered nurses with at least two years of experience in mental health. Exclusion criteria included MHNs without any clinical experience, part-time or temporary status in the two institutions (including those who were not attached to a particular service unit). In addition, authors purposively selected participants with a maximum variation of ethnic and religious backgrounds to enrich the theorization of the data. 

The data collection occurred from October to December 2017. The interviews were conducted primarily in English. The study employed a semi-structured interview guide to collect the data using focus group discussions (FGDs) and individual in-depth interviews. Each participant was involved in only one type of data collection. Forty MHNs contributed to six FGDs (an average of seven people in each group) at the premises of the two participating hospitals. The remaining three participants were interviewed individually. 

The semi-structured interview approach gave enough room for participants to express their knowledge and experiences while helping the interviewer (the first author) to keep the discussion within the scope of the research objectives [29]. The interview topics included: how nurses applied health literacy in their practice in terms of interactions with patients and their relatives, how specific cultural and religious beliefs and practices influenced their interactions with patients and their relatives, and how such interactions affected health literacy practices, such as assessing and dealing with incidence of low health literacy. 

On average, the FGDs lasted about two hours and the individual interviews lasted for 40 minutes. All the interviews were audio-recorded digitally with the help of a research assistant. During the data collection, the first author wrote periodic reflective field notes and insights (memos) about what was learned during the data collection process. 

### 2.3. Data Analysis 

All the audio-recordings were transcribed verbatim. The field notes (memos) that were taken were added to the transcripts before the analysis. The audiotapes were listened to repeatedly to ensure accuracy and to rule-out any missing words or unclear wordings. Conventional content analysis was used to analyze the data to help describe and categories the information into meaningful classes [30]. 

The conventional content analysis involved open (inductive) coding and creating main categories and sub-categories [30]. After open-coding, subcategories were identified and grouped as generic categories, and then consolidated into themes based on their underlining meaning [29,30]. Each category was named using descriptors representing content characteristics. NVivo software for Windows version 11 was used for data storage, organization, and to support analysis.

### 2.4. Trustworthiness 

The authors established four criteria to ensure trustworthiness: credibility, conformability, dependability, and transferability [31]. Credibility was established during the analysis process with the use of two independent researchers reading each transcript to grasp the content and capture the essential features of participants’ responses. A coding book was generated, and comparisons between researchers’ codes occurred. When disagreements occurred, they were resolved through discussion and re-examination of data to ensure conformability. The use of FGDs and in-depth interviews also helped to triangulate different kinds of data to enrich the dependability of the findings [31]. Further, conformability was assured through representative excerpts of raw data in this paper, which help readers make connections between the data and results, and assess transferability to their settings or groups. Finally, results were compared to the existing literature for extrapolation of insights and to raise new questions for future research. 

### 2.5. Ethical Considerations

To ensure anonymity, the two chosen hospitals were labeled as Hospitals A & B. The Human Subject Ethics Sub-Committee of Hong Kong Polytechnic University approved the study protocol (Reference Number: HSEARS20171011006). Permission and consent of the two hospital directors were also sought before commencing the study. Detailed information about the study was provided to the participants, and their written consent was attained. To ensure the anonymity of participants, their identities remained accessible only to the research team, and all dissemination of their information identifies them only by a designated letter, their sex, and staff position. 

## 3. Results 

Forty-three MHNs participated in the study. In brief, 13 male and 30 female from ages 20 to 50 years old participated. In terms of education, 28 had a diploma, 13 had a bachelor’s degree, and two had a master’s degree. Further, they reported considerable participant diversity in years of nursing experience, nursing rank, and ethnicity. For the latter, participants identified themselves as belonging to six different ethnic groups. They worked in various units ranging from out-patient departments to administration departments. Table 1 summarizes the characteristics of the participants.

The findings from the study are broadly grouped under three themes: (1) cultural beliefs direct MHNs’ practice; (2) language, social and institutional factors threaten health literacy practice; and (3) health literacy practice must be negotiated within a cultural context. Findings suggest that although the nurses acknowledged the importance of health literacy associated with patients’ health outcomes, its practice was uncommon. This was strongly attributed to patients’ dependence on cultural practices and beliefs used in the interpretation and treatment of their health problems; sometimes shared among nurses. 

### 3.1. Cultural Beliefs Direct MHNs’ Practice 

Nurses reported that they commonly drew on local cultural beliefs in their practices. The majority of participants described mental illness as rooted in spiritual causes, not biomedical or “physical” ones; some of which they shared with their patients: 

*Ghanaians believe* [that mental illness] *is a curse, sin or wrongdoing against a god by their ancestors, family or themselves for which they are being punished. … A wrong deed against someone is believed to result in mental illness* (Informant D, male, Senior Staff Nurse).

*Yes, some conditions can be seen as spiritual. You just can’t explain the pathophysiology of it. We don’t do CT or MRI scans here, so we can’t explain the causes. …So what we can’t explain physically, we only assume it is caused spiritually, especially in puerperal psychosis, epilepsy, and dementia* (Informant F, female, Senior Nursing Officer, Ward in-charge).

Spiritual explanations appeared to be contradictory to the established nursing knowledge in “pathophysiology” that MHNs were taught in nursing schools. Instead of relying on this knowledge, the participants tended to draw on their culture to make judgments about: Which treatment was the “best” for patients? Who was considered as “ill”? The following quotation illustrates this idea:

*I sometimes ask patients to consider spiritual care because when the condition is spiritual, and [I think] the person can get genuine spiritual healing’. It works because it has to do with the ‘brain and how you think. If the person feels it is spiritual, yes, I will refer them to go see herbalist because they have the rights to choose* (Informant G, female, Senior Staff Nurse)

*Yes, sometimes my cultural background influences the way I educate patients. In my community and even in the church, we reprehend addiction to alcohol. Because of this, I don’t even regard drunkards as sick people. …I usually avoid the substance abuse patients and focus all my attention on those with other conditions* (Informant H, male, nursing officer)

Participants’ knowledge of patients’ cultural beliefs often created challenges for patient education in prescribed medical management plans: 

*Most people do not know about mental health. …It does affect our communication with them. Many are fixed on their beliefs, so it is difficult to get them to trust the orthodox medical practice* (Informant I, male, Senior Staff Nurse). 

*Even the well-educated people, who have strong religious stands, also think mental disorders are caused spiritually. Because of that, I always try to find out the patients’ belief system before educating or explaining issues to them. …But if [the patient] has a strong belief in spirituality, I know I won’t get anywhere with my efforts* (Informant J, female, senior staff nurse).

Consequently, some participants felt helpless in adopting health literacy techniques, and considered their efforts to apply health literacy practice as “difficult,” complaining that it was a “waste of time” to exercise health literacy practice: 

*I sometimes take time to interact and educate patients. …They will nod their head in agreement, but as soon as they leave the hospital, they disregard all the information. …It is just a waste of time educating Ghanaians about mental health issues* (Informant K, male, senior staff nurse).

Some participants reported that there was a lack of public interest in learning medical information about mental health problems. In this regard, participants expressed that, in general, patients were not ready to accept their mental health issues or develop their health literacy in this area, as compared to practices in their physical health: 

*Our patients and the entire Ghanaians do not show any interest in learning about mental health issues. …Sometimes they will have adequate health literacy and know-how to search for information on their physical health problems but they don’t do the same for mental health. …To Ghanaians, they believe, if they don’t abuse drugs, don’t offend any witch, or don’t have bad spirits in their family, mental health is not a concern* (Informant A, female, Senior Staff Nurse).

Nursing practices were reported to be overshadowed by cultural beliefs and practices. Most of the participants reported that they lacked conscious efforts to assess health literacy in their clinical practice. Some participants also reported that their attitudes negated their intention to assess patients’ health literacy prior to health education or nursing care: 

*By the definition of health literacy and my understanding of the term, I don’t think, as mental health nurses, we do assess a patient’s level of health literacy in our hospital. Most of the time we don’t evaluate them but just go ahead with our health education …We don’t take a step further to know their health literacy level, or whether they genuinely understand their information by asking them questions about what we taught them ... we just go ahead with our nursing care* (Informant H, Male, nursing officer). 

Some participants acknowledged that, to some extent, patients’ health literacy was related to the insufficiency of nurses’ knowledge. For example, some reported that they lacked updated information on health issues and/or health education skills: 


*Some nurses eschew health education because they know little (Informant F, female, Ward in-charge) *


*Everyone is talking about the health literacy of the patient. What about the health literacy of us (nurses)? Sometimes nurses don’t understand what they read, or the topic at hand and they give wrong information to the patients. …Majority of the patients go home with false information which leads to devastating effects, including readmissions* (Informant C, male, nursing officer)

### 3.2. Language, Social and Institutional Factors Threaten Health Literacy Practice

Participants reported that language differences between themselves and their patients affected their capacity to engage in health literacy practice. The differences resulted in ineffective communication. According to the participants, their patients were not limited to only Ghanaians but extended to neighboring francophone countries (such as the Ivory Coast and Togo): 

*The other issue with health literacy is language differences. In our setting, we deal with patients from different ethnic groups, and as such, they speak different languages. Because most of the mental health nurses here are not bilingual, it becomes difficult when we meet a patient who cannot speak any of the languages we understand. …We sometimes even receive patients from Togo.* (Informant L, female, nursing officer) 

Aside from language issues, other systemic factors may have inhibited participants from adopting health literacy principles. Specifically, the participants reported that a huge number of patients in hospitals left them limited time for effective health literacy application in terms of assessment and addressing related issues:

*In the current service delivery model, it will be difficult to assess patients’ health literacy or even talk to them. There are always several patients waiting for you. …The time allocated for each patient is limited, so the only thing we can do is to reassure and inform them of the diagnosis and medication* (Informant C, male, nursing officer).

In addition, the lack of materials for promoting health literacy, such as educational and illustrative materials, was noted. Many participants did not have access to any materials for promoting health literacy: 

*‘Talking therapy’. That is what we use. We have no written materials for patient teaching, so we rely on our knowledge of mental health conditions to educate patients* (Informant M, female, nursing officer, Ward in-charge). 

*We don’t have any resources for these patients. We have learned a lot in school, but we can’t apply them in practice due to lack of resources. We have been on countless demonstrations advocating for drugs, educational materials, and improved services, but all fell on deaf ears.* (Informant N, female, senior staff nurse).

Further, they lacked other institutional resources, such as policies reflecting a good understanding of the health literacy issues in mental health: 

*The people formulating the policies do not even know or have adequate mental health literacy. So, they don’t see the importance of mental health as such they won’t formulate policies that will improve mental health. Everybody sees mental problems as a spiritual issue, so little attention is paid to it…The government has neglected the mental health sector* (Informant H, male, nursing officer).

One participant asserted that, without good access to mental health services in Ghana, patients had little choice but to rely on spiritual or “traditional leaders” for guidance:

*When people in the northern part of Ghana begin to see behavioral changes with their relatives, and they often do not know what to do since there is no psychiatric hospital in those places. ...Due to the long distance to the south, they mostly find solace in what the religious personnel and traditional leaders say* (Informant P, female, Ward in-charge).

Finally, social factors that threatened health literacy of mental health appeared to extend into the community. Particularly, some nurses reported that stigmatization was strong in Ghanaian society and this discouraged citizens to talk about mental health. For instance, one participant reported that she was refused to deliver a speech about mental health in a church congregation.

*There was a time we were going to a church to give education on mental health. They decided they were no longer giving us their time if we were not going to talk about the physical illness...* (Informant Q, male, nursing officer).

### 3.3. Health Literacy Practice Must Be Negotiated within a Cultural Context

Some participants expressed that mental health problems were best addressed by taking cultural beliefs into account:

*If you are a nurse and you are going to interact with a community, you must know their beliefs, taboos, values and all those things …We will know which issues to discuss, and what they will also be able to understand better. …A nurse cannot just get up and start informing them that this cultural practice or belief is not good* (Informant R, female, Nurse Manager).

*Until we, as mental health nurses, learn ways to help people understand [mental health issues] through health education. Nothing will be resolved. …We can’t solve the problems due to their low health literacy if we don’t deal with their cultural ideologies in mental health* (Informant N, female, Senior Staff Nurse)**. **

To facilitate effective health literacy, some participants emphasized the importance of involving opinion leaders in the discussion of mental health issues and its relation to cultural beliefs. To them, patients’ communities (led by opinion leaders) and social networks were the major influence on people’s behavior, and this explained why many Ghanaians often disregarded information from health professionals. Informal opinion leaders often contributed to patients’ behavior—some whom were unwilling to obtain, process and understand health information provided by healthcare providers: 

*With these cultural issues, nurses need to go into these communities, talk to the opinion leaders, household heads, and chiefs to learn more about their culture. …It is only when nurses are able to convince these leaders on the negative influence of certain cultural beliefs on the health of the community members that we can solve these issues of spirituality and mental illness… *(Informant R, female, Nurse Manager).

Some participants highlighted the importance of finding appropriate ways of integrating mental health practices to cultural beliefs:

*It will be of great help if we learn to integrate spirituality into our nursing practice. …If a patient is a Christian and believes praying will help them recover faster, we should be able to incorporate that belief into our care for him/her and allow the person to pray…Same applies if the patient is a traditionalist. … I’ll ensure my cultural beliefs do not infringe on that of the patients when I interact with them. I’ll rather encourage the patient to do whatever he/she believes* (Informant G, female, Senior Staff Nurse).

A need for more conformity to patients’ and their families’ cultural/beliefs systems was reported by the participants. Some participants admitted that their health literacy skills were inadequate. Therefore, they reported that possession of both health literacy and cultural competency skills would be critical to enhance their nursing practice:

*…Not just cultural knowledge but also health literacy skills. ...even with my level of education and training, I don’t know much about health literacy. …And I think it will be best if nurses, especially mental health nurses have these two skills to help us address these issues of low knowledge which has resulted in negative opinions and stigma towards mental illness in our country* (Informant R, female, Nurse Manager). 

## 4. Discussion

This study examined how local contexts of cultural precepts and language are connected with mental health nurses’ (MHNs) practice and their understanding of patients’ health literacy in Ghana. The findings indicated that while participants acknowledged the significance of health literacy in mental health education and nursing, its practice was challenging because in many instances, cultural beliefs overshadowed other ideologies (i.e., personal or medical) in their interactions with patients. Assessment of patients’ health literacy level and transmission of health information with the consideration of patients’ health literacy level appeared relegated to a secondary role due to the limitation of resources (i.e., time, space, and materials), organizational issues (i.e., policy) and linguistic diversity. Therefore, nurses’ health literacy practice could not be disconnected from their understanding of patients’ health literacy and the prevailing cultural practices. 

The findings of the current study are in line with those of previous research in Ghana, which posits that the public often perceived mental illness as the consequence of spiritual punishment or sociocultural malpractices rather than genetic or biological distortion [5,6]. Spirituality and cultural practices are usually used as the explanation for the conditions or ailments that are challenging to deal with or not clearly understood [5,32]. Some of the participants of the current study agreed with this by asserting that favoritism towards cultural meaning and implications of mental illness affected patients’ use of mental health services, and this prohibited them from adopting mental health assessment and treatment.

In particular, our findings emphasized how cultural elements such as spirituality, social norms, and practices were used in the clinical settings. Therefore, our findings suggest that health literacy practice cannot be reduced to a linear process separate from one’s prevailing social context [33,34]. Indeed, cultural practices and other belief systems such as religion shaped Ghanaians’ attitudes towards health and health literacy. Culture influences how a person communicates, understands, seeks appropriate health services, uses preventive services, and makes informed decisions about health, disability and end-of-life issues [35]. Hence, aligned to previous literature, health literacy is mediated by a person’s cultural experiences of health and disease, education, aging, language, and social interactions; all of which interact with health services [36,37]. 

Nurses as individuals, are connected to different sociocultural facets in society. They have families, religion, personal beliefs/values, languages, and culture [38]. These social structures can affect their judgment in the interactions with patients and their families [39,40]. Examples of this occurred in the current study when the majority of the MHNs expressed a need to uphold biomedical beliefs in their clinical practice yet reported that their cultural beliefs and social norms sometimes overshadowed their nursing training and influenced the way they interacted with patients. 

Another issue of concern had to do with the lack of bilingual or language interpreters and how nurses applied the principles of health literacy to meet the diverse needs of a range of patients all speaking different languages (over 50 different language) in their clinical settings. Issues of language and communication have been previously documented as affecting both health literacy and health system factors [41,42] as they reduce patients’ ability to understand health information fully [43]. As specified by Rudd, “health literacy happens when patients, or anyone on the receiving end of health communication, and providers on the giving end of health communication, truly understand one another” [44]. Communication was hindered by the differences in languages. Although the MHNs adopted some strategies to address the language differences between themselves and their patients in the present study, language difference was not the sole barrier to health literacy practice. Complex socialization processes that hindered how nurses—both as individuals and as a professional group—from understanding and interpreting patients’ needs and conditions was the key cause of miscommunication. 

Current debates and conceptualization about health literacy and its practice are dominated by Western ideologies of health, which may be in conflict with Ghanaian cultural perspectives [5,45]. Therefore, emerging from this study is a gap between Western biomedical practices and the Ghanaian traditional system of healing. Presently, this gap may be a cause of marginalization of health literacy practice as some of the MHNs alluded to in this study. There was evidence to suggest that the MHNs were fully aware of this conflict but were confronted with the challenges of how to effect change in health behavior through biomedical explanations. To reiterate, this was further exacerbated by their own conflicting paradigms of mental health knowledge and of polarization between the biomedical paradigm based on scientific evidence and the traditional healing paradigm based on indigenous knowledge of the Ghanaian culture. 

From the above, MHNs and other health professionals should understand that Western biomedicine in itself is a cultural production [46] and alien to the Ghanaians and other populations in non-Western places population. This realization can help to loosen the tensions between biomedical principles and traditional ideas that affect health literacy practices. These tensions prompt a need to uncover and contest the exclusionary effects of mainstream practices that influence health and care [46]. Perhaps these tensions contributed to some patients and the public preferring medical advice from their relatives or community leaders rather than that given by health professionals. This reliance on advice from familial sources may reinforce the culture of group conformity which influences a community and an individual’s health behavior. Thus, this implies that it may be difficult for individuals to change their behavior without the wider community implementing similar changes [47]. 

To be able to address health literacy issues, there is a need for nurses and other health professionals to understand the indigenous people’s historical beliefs and health practices through consultation with traditional healers and community leaders as was suggested by participants. This will provide nurses with a better understanding of the role that cultural practices and broader social structures play in health literacy and health. Such an approach may open doors to alternative forms of cultural knowledge and practices, and the underlying agency (e.g., resistance) to structural factors that influence patients’ health literacy [33,48]. 

With regard to the practice of health literacy, health systems must continue to offer education and provide resources (e.g. policies) that encourage health professionals to assess, consider, and better integrate social factors, particularly cultural and language elements in clinical practices. To this end, authors recommend culturally oriented educational materials (e.g. educational comic books or brochures), which could be available in various languages to address the issue of language differences to improve knowledge on mental disorders and provide effective ways of giving medical instructions.

## 5. Strengths and Limitations of the Study

To the best of our knowledge, this is the first study to research issues of health literacy in mental health settings among nurses in Ghana. The findings serve as a foundation to tailor interventions that train nurses in areas such as cultural diversity and health literacy to ensure improved patient outcomes. However, the current study did not include the views of patients. The experiences of patients could have enriched the study with an alternative perspective to health literacy practices of the MHNs as they are the recipients of mental health services. 

## 6. Implications for Policy and Practice

The findings of this study have important implications for healthcare institutions and practitioners. The findings support health institutions’ commitment to effective health literacy promotion and training through inclusion of intercultural practices in orthodox health services across the country. The Ministry of Health should provide mental health practitioners with culturally sensitive educational materials and the necessary support so that the practitioners could promote mental health literacy in communities and clinical settings. The participants in this study proposed that integration of the traditional healing system and Western medicine can be a possible solution to resolve tensions between the two treatment approaches. The high dependence of Ghanaians on the traditional healing system can serve as the first point of call for primary healthcare [40,46]. Nonetheless, the integration of traditional medicine and Western medicine should be developed in consultation with multiple sectors in the society and different stakeholders in the field and not only the orthodox health professionals. Traditional healers, the public, and spiritual leaders should also play active roles in this action. This will then improve the quality of care, health outcomes, and enhance the mental health literacy of the public.

## 7. Conclusions

Results of this study help to understand why and also how local sociocultural practices tend not to support research into mental illness and care in Ghana, as well as to explain the possible reasons for the limited research in clinical settings among nurses [32]. For the most part, this was attributed to dependence on cultural practices and beliefs in the interpretation and treatment of patients’ health problems. Our findings suggest that a re-conceptualization of cultural impact on health literacy is needed to understand the complex interaction between nurses and patients or nurses and families fully in cultural-rich contexts. We recommend future studies may employ a multi-theoretical lens (i.e. mixed methods) to uncover health inequalities, poor health outcomes, and stereotypes that are caused by limited health literacy practice. 

## Figures and Tables

**Table 1 ijerph-16-03589-t001:** Characteristics of the mental health nurses involved in the study.

Characteristic		Frequency N = 43	Percentage
Hospital	Hospital A	25	58.1
	Hospital B	18	41.9
Gender	Male	13	30.2
	Female	30	69.8
Age	20-30 years	12	27.9
	31-40 years	26	60.5
	41-50 years	5	11.6
Ethnicity	Akan	23	53.5
	Ewe	7	16.3
	Ga	7	16.3
	Dagomba	1	2.3
	Guan	2	4.7
	Dagaati	3	7.0
Departments	Out-patient department	6	14.0
	Female ward	19	44.2
	Male ward	9	20.9
	Rehabilitation Unit	5	11.6
	Nursing Administration	4	9.3
Level of Nursing education	Diploma	28	65.1
	Bachelor’s degree	13	30.2
	Master’s degree or above	2	4.7
Years of Nursing Experience	≤10 years	11	25.6
	11-20 years	22	51.2
	21-30 years	7	16.3
	≥30	3	7.0
Nursing Rank	Staff Nurse	1	2.3
	Senior Staff Nurse	16	37.2
	Nursing Officer	17	39.5
	Senior Nursing Officer	4	9.3
	Principal Nursing Officer	1	2.3
	Nurse Manager	4	9.3

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
