# Peer review of "“I Sometimes Ask Patients to Consider Spiritual Care”: Health Literacy and Culture in Mental Health Nursing Practice"

_ijerph, 2019, doi:10.3390/ijerph16193589_

Round 1
Reviewer 1 Report
The article is focused on interesting and important aspect of health care. The structure of the paper respects the journal's format. The organization of paper is consistent and logic. The methodology is strong and results are presented in the proper scientific manner. Coverage of existing literature is adequate.
I have few suggestions how to improve your article:
Please add in the "Methods" section information regarding the time of the study (when your study was performed?). It would be also interesting for readers to see the questions and scenario for focus groups and in-depth interviews. Maybe you can present/attach these files in Supplementary materials.The relevance of this article could be increased by including specific recommendations. The implications and policy recommendations defined in "Discussion" section, should be more specific and elaborated.
Author Response
Response to Reviewer 1
We appreciate your precious comments as these are helpful to improve this manuscript. We have made the following corresponding amendment based on your comments.

Reviewer 2 Report
I would like to congratulate the authors on a well-prepared and interresting article. I have a special interest in culture, spirituality and mental wellbeing and enjoyed reading the document.
However, I feel that the Lit Review should include a short description of the various cultures, languages, etc in Ghana, so the reader has a better understanding of various challenges as presented in the article.
Line 61: An explanation as to why nurses are not equipped as well as possible solutions to this challenge.
Line 382; 383; 391; 393 'western' spelt with capital "W".
Author Response
Response to Reviewer 2
We appreciate your comments as these are helpful to improve this manuscript. We have made the following corresponding amendment based on your comments in the attached file. Thank you
